# Facile Synthesis of Cu-Doped ZnO Nanoparticles for the Enhanced Photocatalytic Disinfection of Bacteria and Fungi

**DOI:** 10.3390/molecules28207232

**Published:** 2023-10-23

**Authors:** Ruichun Nan, Shurui Liu, Mengwan Zhai, Mengzhen Zhu, Xiaodong Sun, Yisong Chen, Qiangqiang Pang, Jingtao Zhang

**Affiliations:** 1The Institute of Vegetables, Hainan Academy of Agricultural Sciences, Key Laboratory of Vegetable Biology of Hainan Province, Haikou 571100, China; 2School of Food and Bioengineering, College of Tobacco Science and Engineering, Zhengzhou University of Light Industry, Zhengzhou 450002, China; 3Luohe Weilong Biotechnology Co., Ltd., Luohe 462000, China

**Keywords:** Cu-doping, nanoparticles, semiconductors, antifungal activity, safety assessment, Zinc oxide

## Abstract

In this study, Cu-doped ZnO was prepared via the facile one-pot solvothermal approach. The structure and composition of the synthesized samples were characterized by XRD (X-ray diffraction), TEM (transmission electron microscopy), and XPS (X-ray photoelectron spectroscopy) analyses, revealing that the synthesized samples consisted of Cu-doped ZnO nanoparticles. Ultraviolet–visible (UV-vis) spectroscopy analysis showed that Cu-doping significantly improves the visible light absorption properties of ZnO. The photocatalytic capacity of the synthesized samples was tested via the disinfection of *Escherichia coli*, with the Cu-ZnO presenting enhanced disinfection compared to pure ZnO. Of the synthesized materials, 7% Cu-ZnO exhibited the best photocatalytic performance, for which the size was ~9 nm. The photocurrent density of the 7% Cu-ZnO samples was also significantly higher than that of pure ZnO. The antifungal activity for 7% Cu-ZnO was also tested on the pathogenic fungi of *Fusarium graminearum*. The macroconidia of *F. graminearum* was treated with 7% Cu-ZnO photocatalyst for 5 h, resulting in a three order of magnitude reduction at a concentration of 10^5^ CFU/mL. Fluorescence staining tests were used to verify the survival of macroconidia before and after photocatalytic treatment. ICP-MS was used to confirm that Cu-ZnO met national standards for cu ion precipitation, indicating that Cu-ZnO are environmentally friendly materials.

## 1. Introduction

A continuous deterioration in water quality due to pathogenic pollution has raised significant concerns among researchers worldwide [1,2,3]. Statistics indicate that water-borne diseases caused by water pollution affect millions of people globally and cause about 14,300 deaths every year [4]. The invention of efficient sterilization technology is of great significance for water purification and reduction in the occurrence of water-borne diseases. Thus far, traditional sterilization technologies such as ultraviolet sterilization [5] and antibiotic sterilization [6] have been unable to meet requirements due to drawbacks, including high costs, secondary pollution, and bacterial resistance. Compared with these traditional sterilization methods, photocatalysis-based disinfection technology exhibits tremendous prospects for application in the field of sewage treatment as it is environmentally friendly and causes no secondary pollution [7,8].

The excellent photocatalytic performance of metal oxides makes them an important research object in the field of sewage treatment. A series of metal oxides including TiO_2_, ZnO, Cu_2_O, and WO_3_ have been found to have photocatalytic potential [9,10,11]. The characteristics of non-toxicity, biocompatibility, thermal stability, photostability, and high UV absorption make ZnO a promising photocatalyst [12,13]. However, its low light absorption capacity and high electron–hole recombination rate limit the application of pure ZnO in the field of photocatalysis [14].

The modification of materials is an important means of widening this narrow range. Transition metals such as Ag [15], Pd [16], Mn [17], and Cu [18], due to their valence states, are easy to change; outer electrons can easily be removed, and they can easily acquire electrons on the d orbital, providing them with redox properties [19]. Hence, they are widely used as modifiers of photocatalysts to overcome the aforementioned deficiencies. Among the transition metals, Cu is the most prominent; on the one hand, it has a certain bactericidal effect, which enables it to create a synergistic effect with photocatalytic fungicides [20]. On the other hand, it has the lowest cost and good biocompatibility compared with other transition metals [21].

The Cu-doping of ZnO is a good strategy for improving its photocatalytic efficiency. In this study, Cu-doped ZnO was prepared using the solvothermal method. The solvothermal process refers to the process of decomposition or chemical reaction between precursors induced by the presence of a solvent in a closed reaction vessel at a temperature higher than the boiling point of the solvent [22]. Depending on the experimental conditions (pressure and temperature), solvothermal systems can be heterogeneous, subcritical, or supercritical. Wojnarowicz J [23] proposed that the solvothermal method is an environmentally friendly method, and it is recognized as a green chemical method. The main advantages include high speed, good uniformity, and high purity. The experimental results show that the hydrothermal synthesis of ZnO nanomaterials has great potential and can obtain a wide range of product morphology, from quantum dots to core–shell structures and layered structures. In addition, Cu/ZnO nanoparticles were prepared by Fu [24] using the sol–gel method, whereby the maximum degradable rate of methyl orange reached 88% when the initial concentration was 20 mg/L during a photodegradation test of nanomaterials. Kuriakose et al. synthesized Cu/ZnO using the wet chemical method and examined the mechanism underlying its excellent photocatalytic performance; the authors attributed this performance to the formation of electronic defects, as well as the formation of a heterojunction between Cu and ZnO, promoting carrier separation [25]. Cu-doped ZnO nanoparticles were prepared by Khan [26] using the coprecipitation method, while the antibacterial activity of different strains was studied by means of the agar diffusion method. The results indicate that synthesized Cu-doped ZnO nanoparticles have excellent antibacterial activity against *Escherichia coli* and *Bacillus subtilis*, with inhibitory regions of 31 ± 0.4 mm and 27 ± 0.6 mm, respectively. Subsequently, Cu-doped ZnO was synthesized by Rashid [27] using the coprecipitation method. The author highlighted that the disinfection effect of ZnO on Gram-positive bacteria was significantly different (*p* < 0.05) before and after Cu-doping. By observing the structural changes before and after doping, the reason for the increase in the photocatalytic activity was found to probably be the transformation of nanoclusters into nanorods when Cu was doped into ZnO. Thus, ZnO doped with Cu produces good photocatalytic activity.

In the present study, Cu-doped ZnO was synthesized with the aim of achieving an enhanced photocatalytic effect. The characterization of the structural morphology, element composition, and valence distribution of the nanomaterials was proved using X-ray diffraction (XRD), transmission electron microscopy (STEM), and X-ray photoelectron spectroscopy (XPS) analyses. The carrier separation performance of Cu-ZnO was illustrated using electrochemical impedance spectroscopy (EIS) and isothermal transformation (I-T) curves at an electrochemical workstation. The light absorption performance of Cu-ZnO was characterized via ultraviolet–visible (UV-vis) spectroscopy. The nanomaterial’s actual photocatalytic efficiency was determined by measuring the effect of Cu-ZnO on the Gram-negative bacteria *E. coli* and the pathogenic fungus *F. graminearum*. Finally, the safety assessment for Cu-ZnO of the release of Cu ions was tested using ICP-MS. Herein, we provide a promising environmentally friendly and visible light excitation catalyst as an alternative method to disinfect bacteria and agricultural pathogenic fungi.

## 2. Results and Discussion

### 2.1. Material Characterization

The full XRD spectrum and the local amplification spectrum of Cu-ZnO with different proportions at a diffraction angle of 20~80° are displayed in Figure 1a and Figure 1b, respectively. The peaks corresponding to 31.7°, 34.4°, 36.2°, 47.5°, 56.6°, 64.8°, and 66.3° are the characteristic diffraction peaks of hexagonal wurtzite ZnO, reflecting the crystal planes (100), (002), (101), and (102) [28,29]. The calculated crystallite sizes from the XRD results were 8.23 nm, 8.11 nm, 8.56 nm, 8.93 nm, 9.13 nm, 9.2 nm, and 9.23 nm for the ZnO, 1% Cu-ZnO, 3% Cu-ZnO, 5% Cu-ZnO, 7% Cu-ZnO, 10% Cu-ZnO, and 20% Cu-ZnO, respectively. The positions of the main peaks were not seen to change significantly with the increase in the Cu-doping amount (Figure 1a); no impurity peaks can be seen in the detection range of the XRD method. We speculate that the Cu may exist in the form of ions. Figure 1b shows that the spectrum after the position of the main peak is locally enlarged. We can see that the main peak tends to shift to the right and that the height of the peak decreases with the increase in the doping amount, which proves that the Cu was doped into the ZnO [30,31], leading us to tentatively speculate that the Cu was doped into the ZnO in the form of Cu ions. We tested the Cu content in the samples, which are shown in Table 1. The wt.% Cu content in the samples was 0.74%, 2.07%, 3.23%, 4.33%, 6.45%, and 11.62% for the 1% Cu-ZnO, 3% Cu-ZnO, 5% Cu-ZnO, 7% Cu-ZnO, 10% Cu-ZnO, and 20% Cu-ZnO, respectively.

The TEM results for the Cu-ZnO nanocomposites are shown in Figure 2a. No Cu single particles can be found on the surface of the ZnO nanoparticles. The particle size ranges from 3.7 to 16.8 nm, which counted from 200 nanoparticles of the TEM images. In Figure 2b, the lattice distance of the exposed crystal plane is measured as 0.28 nm and 0.26 nm, corresponding, respectively, to the (100) crystal plane and the (002) crystal plane of the card of the characteristic diffraction peak of ZnO (JCPDS no. 36-1451). No lattice stripe of Cu is found. Observing the HRTEM of Cu-ZnO, it can also be seen that only ZnO particles appear in the material, and there is no single substance of Cu, which is consistent with the observation results from the XRD spectrum analysis. Scanning TEM, as shown in Figure 3, was used to study the microstructure and element distribution of the synthetic materials. It can be seen from the figure that the elements Cu, O, and Zn are evenly dispersed in the synthetic material.

The XPS spectrum and fitting analysis results are shown in Figure 4. The full XPS spectrum in Figure 4a shows that carbon (C 1s), zinc (Zn 2p), copper (Cu 2p), and oxygen (O1s) exist in the composite. The XPS spectrum of O1s was amplified and analyzed, as shown in Figure 4b. The peaks of 530.2 eV and 531.8 eV were fitted and identified as an oxygen lattice in the Zn-O [32] and hydroxyl (O-H) [33] adsorbed on the material surface, respectively. The peaks of Zn 2p were fitted as 1021.2 eV and 1044.4 eV, respectively. In Figure 4c, they correspond to Zn 2p3/2 and Zn 2p1/2, respectively, indicating that the Zn element in the prepared composite exists as Zn^2+^, which proves that ZnO is indeed present [34], which is consistent with the previous results of XRD and TEM analysis. Furthermore, 934.9 eV and 954.7 eV correspond to the fitting peaks of Cu, representing Cu^2+^ 2p3/2 and Cu^2+^ 2p1/2, respectively. The two satellite peaks can be observed in Figure 4d, which proves that the synthesized composite contains Cu^2+^ [35]; this is consistent with our previous speculation on the existing form of Cu in XRD and TEM. Thus, Cu was doped in ZnO in the form of ions.

Figure 5 shows the UV-vis light absorption capacity of the samples. Pure ZnO has intrinsic light absorption characteristics. The samples containing Cu-ZnO nanoparticles clearly show a shift of the light absorbance into the visible light range. In addition, with the increase in the Cu atomic ratio, the visible light absorbance of the Cu-ZnO samples is gradually enhanced. Table 2 shows the surface area of the synthesized samples via the BET test. The surface area of the synthesized samples was 49.1–59.1 m^2^/g.

### 2.2. Photocatalytic Sterilization Performance

The bactericidal efficiency of Cu-ZnO with different doping ratios under visible light conditions is shown in Figure 6. The survival rate of *E. coli* was reduced to 92.67% after 150 min of visible light irradiation and without any additional materials. The decrease in the number of bacteria was greater than under dark conditions, indicating that this decline was caused by light damage [36]. Following the addition of pure ZnO, the survival rate of *E. coli* was reduced to about 87.82% after 150 min of visible light irradiation, proving that ZnO offers photocatalytic performance [37]. The photocatalytic disinfection performance shows clear improvement after doping with Cu ions. The 7% Cu-ZnO material presents the best sterilization performance, reducing the number of bacteria by seven orders of magnitude after 120 min of 30 mW/cm^2^ light intensity. The 10% Cu-ZnO material reduced the number of bacteria by seven orders of magnitude after 150 min, while the 3%, 5%, and 20% Cu-ZnO all reduced *E. coli* by about five orders of magnitude. In all of the synthesized samples, 7% Cu-ZnO exhibited the best photocatalytic performance. Thus, we tested this as one of the most important samples of TEM, XPS, and EIS.

The bactericidal property of the 7% Cu-ZnO bacterial solution was further explored under dark conditions, whereby the *E. coli* survival rate reached 43.09% after 120 min of contact, which may have been due to the bactericidal effect of the Cu itself [38]. The height of the peaks is related to the separation of electrons and holes, whereby the lower the peak, the higher the separation, and the corresponding photocatalytic efficiency also increases. Figure 6 shows that the peak of 7% Cu-ZnO has the lowest height, followed (in descending order) by 10%, 5%, 20%, 3%, and 1% Cu-ZnO; the corresponding electron–hole separation efficiency is proportional to the sterilization performance [39,40].

The electron impedance spectrum (EIS) shown in Figure 7a reflects the electron–hole recombination efficiency of the materials. The size of the radius is used to reflect the recombination rate of electrons and holes in the high-frequency region, whereby the larger the radius, the higher the electron–hole recombination rate. The electron–hole recombination rate of 7% Cu-ZnO is significantly lower than that of pure ZnO. The number of photoelectrons produced by the nanomaterials under light and dark conditions is shown by the transient photocurrent response spectrum. Figure 7b shows that under light conditions, the photocurrent density of the 7% Cu-ZnO composite is stable at about 0.8 μA cm^−2^, which is much higher than the photocurrent density of pure ZnO. Under dark conditions, the photocurrent density of 7% Cu-ZnO tends to be similar to that of pure ZnO, which conclusively shows that 7% Cu-ZnO produces photoelectrons under light conditions and that its efficiency is much higher than that of pure ZnO. Thus, the high photocurrent density of 7% Cu-ZnO could be due to the high separation efficiency of the electron holes.

### 2.3. Antifungal Performance Test

Given the excellent antibacterial properties of Cu/ZnO, 7% Cu-ZnO was singled out for use as a control performance assay for macroconidia of *F. graminearum*. As can be seen from the fungicidal curves in Figure 8, the survival of macroconidia was proportional to the duration of the photocatalytic treatment, with the population of macroconidia decreasing by three orders of magnitude within 5 h of photocatalytic treatment with 7% Cu-ZnO. In contrast, under dark conditions, 7% Cu-ZnO had almost no effect on the activity of macroconidia of *F. graminearum*.

### 2.4. Fluorescence Staining Tests

Figure 9a and Figure 9b, respectively, present the phase difference images and fluorescence staining images of macroconidia of *F. graminearum* prior to photocatalytic treatment. Before the treatment, the spores were morphologically intact with a smooth surface, and Figure 9b clearly shows that the spores were almost all stained green, demonstrating their good activity. Figure 9c and Figure 9d, respectively, present the phase difference images and fluorescence staining images of macroconidia of *F. graminearum* spores after photocatalytic treatment. The spores’ morphology exhibited changes following the treatment, in that the cell structure had been destroyed and the spore contents had flown out. Figure 9d further reveals that most of the spores were stained red, representing dead spores [39].

### 2.5. Safety Test

The precipitation degree of Cu ions is an important index to evaluate the safety of Cu-ZnO for use in water source treatment [40]. Figure 10 shows the release of Cu ions into water [41], taking 20% Cu-ZnO oxide as an example (tested using ICP-MS). The results show that the solubility of Cu ions was 0.07 mg/L before visible light irradiation, which may be due to the dissolution of a small number of copper clusters during ultrasonic dispersion. These dissolved Cu ions may have had an impact on the number of *E. coli* in the sterilization experiment, which supports our previous conclusion that the material also has a bactericidal effect under dark conditions due to the Cu ions. With the increase in illumination time, the solubility of the Cu ions in the composite gradually increased, reaching 0.13 mg/L at 150 min. The hygienic standard for drinking water [42] (GB5749-2006) issued by the UK Ministry of Health states that the detection limit for Cu ion concentrations in drinking water is 1.0 mg/L [43]; the maximum Cu ion solubility of 20% Cu-ZnO nanoparticles in this experiment was significantly less than 1.0 mg/L. Therefore, the concentration of Cu ions dissolved in water with the Cu-ZnO nanoparticles synthesized in this experiment was far below the safety threshold, and the potential for impact on the human body and the environment is therefore negligible.

## 3. Material and Methods

### 3.1. Experimental Reagents

Copper chloride (CuCl_2_ • 2H_2_O, 99%), hydrochloric acid (HCl, 37 ± 1%), and absolute ethanol (C_2_H_5_OH, 99.7%) were purchased from Sinopharm Chemical Reagent Co., Ltd. (Shanghai, China). Syringe filters (0.22 μm) were purchased from Shanghai Macklin Biochemical Technology Co., Ltd. (Shanghai, China). Zinc chloride (ZnCl_2_, 99.95%) and sodium hydroxide (NaOH, 97%) were purchased from Shanghai Aladdin Reagent Co., Ltd. (Shanghai, China). Sodium chloride (NaCl, 99.5%), potassium chloride (KCl, 99.5%), potassium dihydrogen phosphate (KH_2_PO_4_, 99%), and disodium hydrogen phosphate (Na_2_HPO_4_ • 12H_2_O, 99%) were purchased from Tianjin FENGCHUAN Chemical Reagent Co., Ltd. (Tianjin, China). All these reagents were of analytical-grade purity. In addition, tryptone was purchased from Beijing Sulaibao Technology Co., Ltd. (Beijing, China), and yeast extract powder and nutrient agar were purchased from Beijing Aoboxing Biological Reagent Co., Ltd. (Beijing, China).

### 3.2. Preparation of Cu-ZnO Nanocomposites

In total, 30 mL measures of absolute ethanol were added to two beakers. A total of 0.15 M (1.252 g) ZnCl_2_ and CuCl_2_ with different molar ratios (Cu: Zn of 0:0.15 mM, 0.0015 mM:0.15 mM, 0.0045 mM:0.15 mM, 0.0075 mM:0.15 mM, 0.0105 mM:0.15 mM, 0.015 mM:0.15 mM, and 0.003 mM:0.15 mM) were added to one beaker, and the mixture was stirred until it was completely dissolved; this was recorded as solution A. A fixed molar ratio of NaOH to ZnCl_2_ and CuCl_2_ of 2:1 was added to another beaker, and this was also stirred until it was completely dissolved; this was recorded as solution B. Then, solution B was added dropwise to solution A and mixed well to obtain solution C. Then, the pH of solution C was adjusted to neutral using hydrochloric acid. Finally, solution C was transferred to a 100 mL reactor and reacted at 80 °C for 2 h. After the reaction, Cu-ZnO nanocomposites with different proportions were obtained after washing (water washing and alcohol washing alternately) and drying; these were recorded as pure ZnO and 1%, 3%, 5%, 7%, 10%, and 20% Cu-ZnO.

### 3.3. Characterization

The nanomaterials’ crystal properties, morphology and structure, and valence distribution were characterized using XRD (D8 Advance, Bruker, Germany), TEM (JEM-2100, JEOL, Tokyo, Japan), and XPS (ESCALAB 250, Thermo Fisher Scientific, Waltham, MA, USA), respectively. The nanomaterials’ photoelectric properties were characterized via UV-vis (U-3900H, Hitachi, Tokyo, Japan), BET (TriStar II Plus 3030, Micromeritics, Norcross, GA, USA) transient photocurrent response (I-T) (CHI660E, Corrtest Instruments, Wuhan, China), and electronic impedance (EIS) (CHI660E, Corrtest Instruments, Wuhan, China).

#### Electrochemical Workstation

The transient photocurrent response (I-T) and electronic impedance (EIS) of the composite were measured using an electrochemical work station. The internal resistance and responsiveness to visible light of the synthetic material were obtained by testing the photoelectric properties of the sample. The specific methods and principles were as follows: 5 mg of the sample was dissolved in 5 mL of absolute ethanol; 1 mL of this solution was extracted ultrasonically mixed with water; and an FTO conductive surface was coated with the mixed liquid to form an area 1 cm^2^ in size. On this 1 cm^2^ area, the working electrode was installed using a platinum sheet electrode as the comparison electrode and Ag/AgCl as the reference electrode; 0.1 M sodium sulfate solution was prepared as the electrolyte for the system, the light source was a 300 W xenon lamp W (ultraviolet light was blocked using a >400 nm filter), and a quartz beaker was used. An ordinary beaker could not be used in this experiment as the intensity of the light would have been weakened to a certain extent.

The amount of photogenerated charge generated by the material under illumination was reflected by the photocurrent curve during different periods. Before the photocurrent test, the light source needed a stability time of 5 min, and following that, every 20 s was taken as a test time (the test was conducted alternately under dark and light conditions, and each test time was 20 s, forming a complete cycle). The light source was blocked with a tin foil baffle to create the dark condition, and for the light condition, the baffle was removed. These two treatments formed a stable current. After five such cycles, the stability and photocatalytic performance of the material was determined. The resistance of the materials to conductive media was determined by AC impedance. The charge transfer ability of the materials, as well as their photocatalytic activity, was indirectly determined by comparing the internal resistance of the different materials.

### 3.4. Disinfection Test

During the disinfection experiment, the concentration of the samples was adjusted to 1 mg/mL at a 10 mL reaction system; *E. coli* K12 with an initial bacterial concentration of 2 × 10^7^ CFU/mL was used as the test strain. In the antifungal test, the initial macroconidium of *F. graminearum* concentration was 2 × 10^5^ CFU/mL. Under visible light with an intensity of 30 mW/cm^2^ and a wavelength range of 420–780 nm, the influence of a different nanomaterial disinfection effect was explored. All tests were repeated three times.

### 3.5. Fluorescence Staining Tests

The survival status of cells before and after treatment with Cu-ZnO nanomaterials was assessed using a fluorescent staining assay. The cell mixture was collected after photo-catalytic treatment with the Cu-ZnO material (the comparison sample was a cell suspension at a concentration of 2 × 10^6^ CFU/mL without any treatment). After washing and centrifugation, the samples were mixed with the fluorescent stain, and the resulting mixture was incubated at 37° for 30 min. Finally, 5 uL each of the untreated and treated *F. graminearum* samples was sampled and observed under a fluorescent phase contrast microscope.

### 3.6. Safety Assessment Tests

Inductively coupled plasma mass spectrometers (ICP-MS) are commonly used to qualitatively and quantitatively analyze trace metal elements in solution. In this experiment, an ICP-MS (ELAN9000, PerkinElmer Life and Analytical Sciences, Shelton, CT, USA) was used to determine the concentration of Cu ions dissolved from the Cu-ZnO nanocomposites during the experiment. This showed whether the solubility of the Cu ions in the composite exceeded the safety threshold, and whether the material could cause harm to the environment or the human body.

## 4. Conclusions

Cu-doped ZnO nanoparticles were successfully prepared via the solvothermal method. The samples as synthesized were characterized using XRD, TEM, and XPS analyses. The characterization results show that the synthesized nanomaterials are Cu-doped ZnO nanomaterials. The photocurrent density of the 7% Cu-ZnO was also significantly enhanced compared to that of pure ZnO. The sterilization experiments proved that the photocatalytic disinfection performance of Cu-ZnO is significantly superior to that of pure ZnO. Among the synthesized materials, 7% Cu-ZnO presented the best photocatalytic sterilization performance, killing almost all bacteria and fungi spore within 120 min. In addition to this, Cu-ZnO are environmentally friendly materials, as demonstrated by ICP-MS testing.

## Figures and Tables

**Figure 1 molecules-28-07232-f001:**
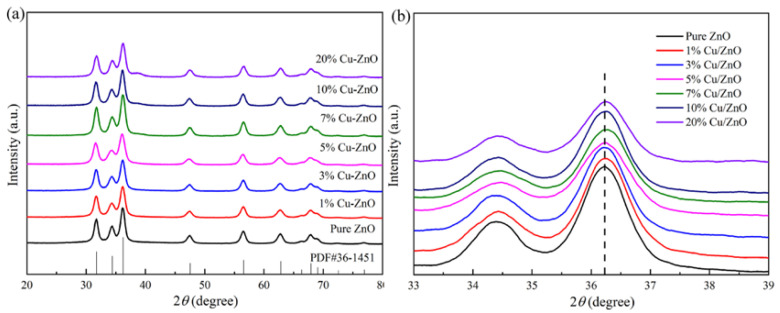
(**a**) XRD patterns and (**b**) local amplification patterns of ZnO and Cu-ZnO with different doping ratios.

**Figure 2 molecules-28-07232-f002:**
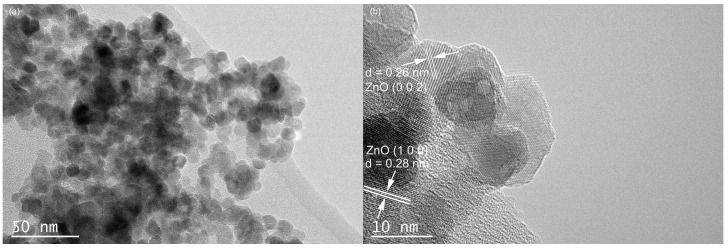
TEM (**a**) and high-resolution spectra (**b**) of 7% Cu-ZnO.

**Figure 3 molecules-28-07232-f003:**
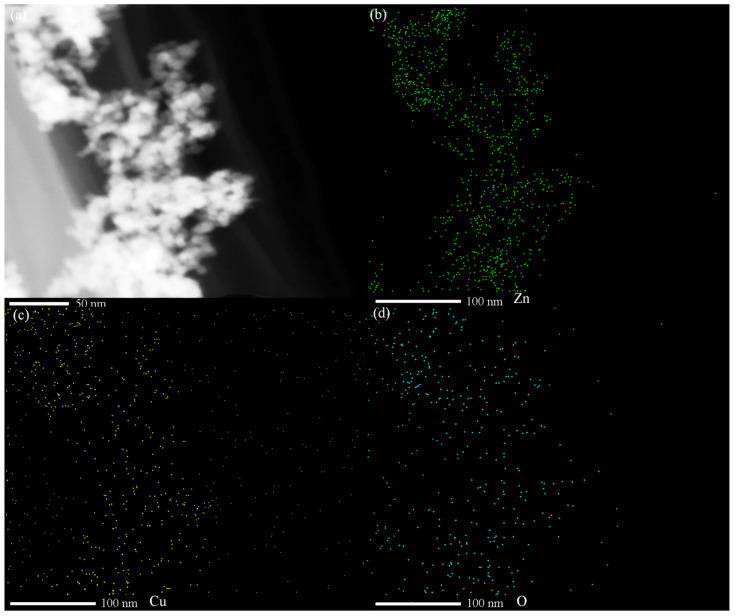
(**a**) STEM spectra of 7% Cu-ZnO samples; STEM mapping images of (**b**) Zn, (**c**) Cu, and (**d**) O. (green: Zn, yellow: Cu, cyan: O).

**Figure 4 molecules-28-07232-f004:**
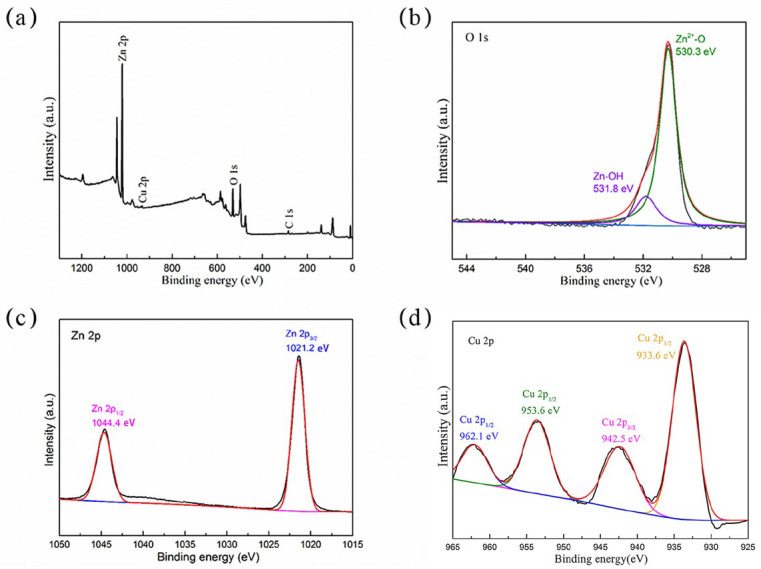
(**a**) XPS full spectrum, (**b**) peak fitting spectrum of O 1s, (**c**) peak fitting spectrum of Zn 2p, and (**d**) peak fitting spectrum of Cu 2p of 7% Cu-ZnO nanomaterial.

**Figure 5 molecules-28-07232-f005:**
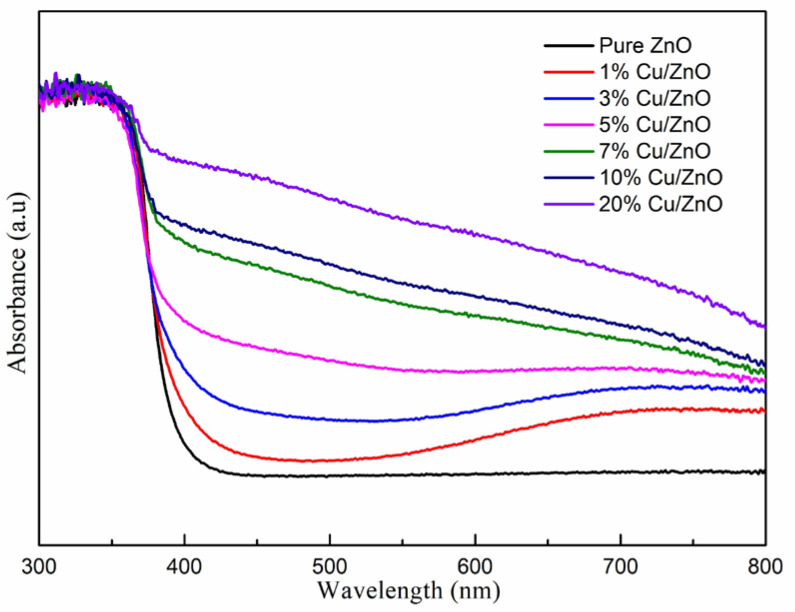
UV-vis absorption spectra of the Cu-ZnO samples.

**Figure 6 molecules-28-07232-f006:**
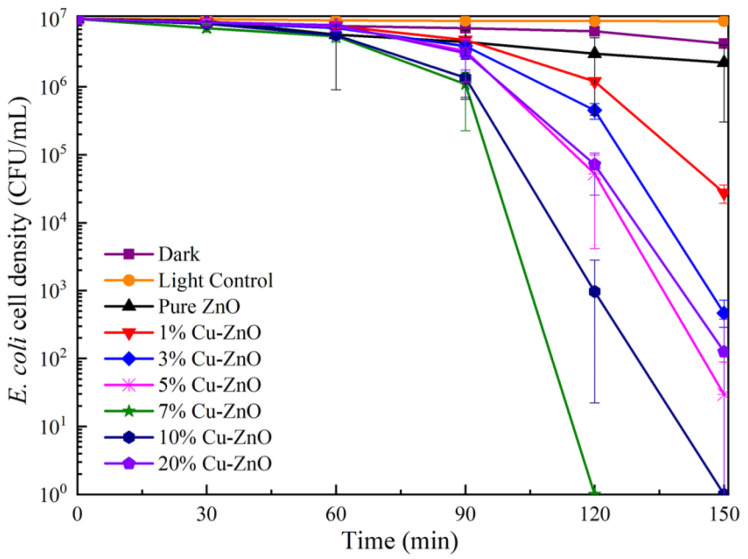
Survival ratio of *E. coli* under photocatalytic using Cu-ZnO samples. The liquid phase volume and the sample mass used for the disinfection experiment were 10 mL and 10 mg, respectively. The data shown were the average values from three experiments.

**Figure 7 molecules-28-07232-f007:**
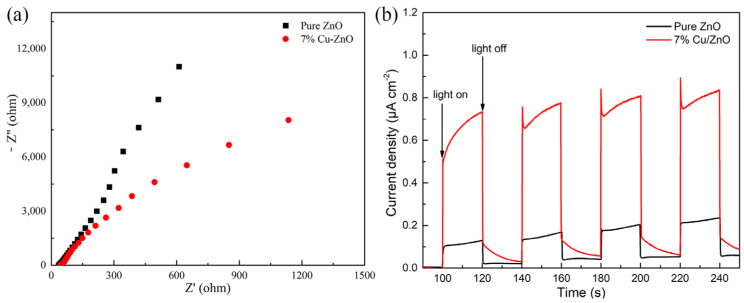
(**a**) EIS spectra and (**b**) I-T spectra of 7% Cu-ZnO samples.

**Figure 8 molecules-28-07232-f008:**
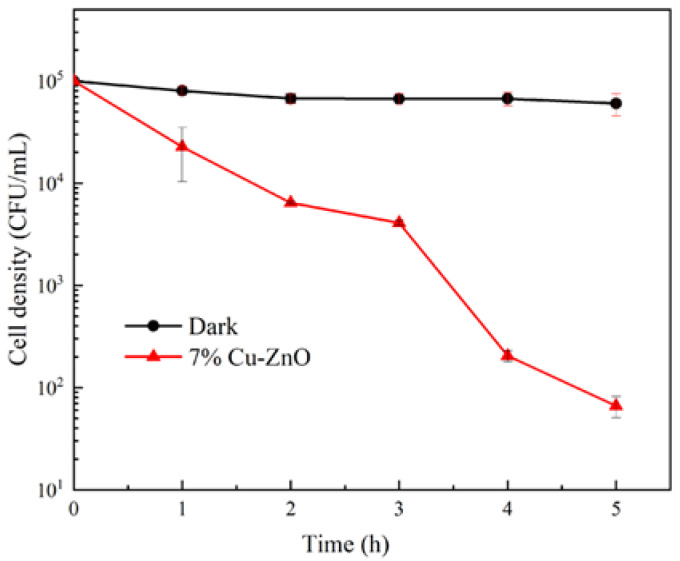
Fungicidal profile of 7% Cu-ZnO against macroconidia of *F. graminearum* under dark or visible light conditions. The liquid phase volume was 10 mL and the mass of 7% Cu-ZnO was 10 mg in this disifection experiment. The data shown were the average values from three experiments.

**Figure 9 molecules-28-07232-f009:**
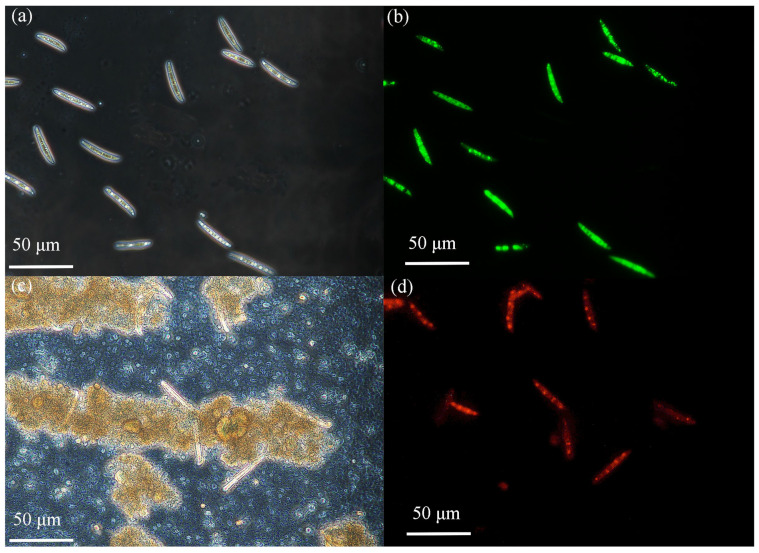
(**a**,**c**) Phase contrast microscope images of the corresponding fluorescent images (**b**,**d**) showing the cell viability of *F. graminearum* macroconidia post 5 h illumination, (**a**,**b**) without photocatalyst or (**c**,**d**) with 7% Cu-ZnO. The cell was stained by CFDA-AM/Propidium Iodide, and the live cells appear green while the dead ones are red in the images.

**Figure 10 molecules-28-07232-f010:**
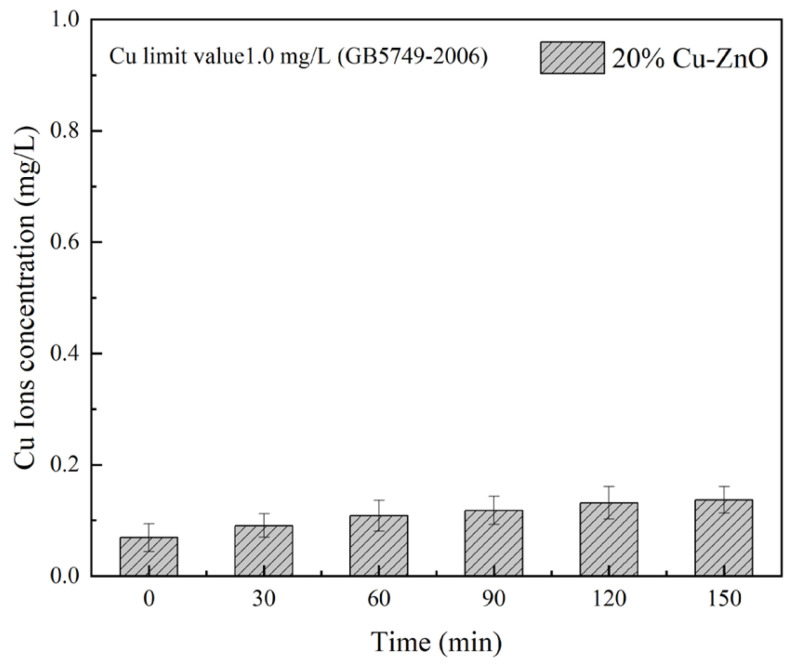
ICP-MS images of 20% Cu-ZnO samples. The liquid phase volume was 50 mL and the mass of 20% Cu-ZnO was 50 mg in this experiment. The data shown were the average values from three experiments.

**Table 1 molecules-28-07232-t001:** ICP-MS test of Cu-doped ZnO nanoparticles with different doping ratios.

S.NO.	Materials Name	wt. (%)
1	1% Cu-ZnO	0.74
2	3% Cu-ZnO	2.07
3	5% Cu-ZnO	3.23
4	7% Cu-ZnO	4.33
5	10% Cu-ZnO	6.45
6	20% Cu-ZnO	11.62

**Table 2 molecules-28-07232-t002:** BET surface area of the synthesized samples.

S.NO.	Materials Name	BET Surface Area (m^2^/g)
1	ZnO	52.3
2	1% Cu-ZnO	54.9
3	3% Cu-ZnO	51.1
4	5% Cu-ZnO	50.5
5	7% Cu-ZnO	49.1
6	10% Cu-ZnO	59.1
7	20% Cu-ZnO	59.1

## Data Availability

The data presented in this study are available on request from the authors.

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
