# Peer review of "Facile Synthesis of Cu-Doped ZnO Nanoparticles for the Enhanced Photocatalytic Disinfection of Bacteria and Fungi"

_molecules, 2023, doi:10.3390/molecules28207232_

Round 1

Reviewer 1 Report

The manuscript deals with the synthesis of Cu/ZnO2 nanoparticles, their characterization, and the application of this material for photocatalytic disinfection of biological material. It was shown that after the addition of Cu to ZnO, the photocatalytic efficiency was significantly improved. The results presented are interesting, but some revisions are needed before the manuscript can be recommended for publication.

The volume of the 0.15 M ZnCl2 solution which was used to synthesize the different Cu/ZnO catalyst should be added to the manuscript (page 8).

What means a “certain amount of flake NaOH” (page 8)?

In the experimental part, the measurement of Mott-Schottky curves was mentioned (page 8). The manuscript does not mention experimental details for this measurement, nor does it show results based on the obtained Mott-Schottky curves. Does the addition of Cu change the valence band maximum and the conduction band minimum edge of ZnO or were new stages inside the band gap generated?

What was the liquid phase volume and the catalyst mass used for the disinfection experiment (page 9)? The reaction condition should be shortly summarized in the caption of the Figures which presenting photocatalytic results.

The conditions for the experiment which was used to determine the Cu content in solution are unclear. What was the catalyst mass and the volume of the solution at this experiment (page 9).

In the manuscript, the nominal Cu content was given. Was this content verified by experimental methods? Can the authors be sure that all the Cu used was ultimately in the catalyst?

The BET surface area of the different photocatalyst should be determined and added to the manuscript.

It was speculated that Cu was doped into the ZnO in the form of ions. EPR measurements might be useful to prove this assumption. Is there also the possibility that small CuO clusters are formed, which are located inside or on the surface of ZnO lattice? The XRD pattern of the 20% Cu/ZnO sample show clearly a small reflection at about 40 ° (Figure 1a). Rietvield analysis of the XRD pattern with higher Cu loading might help here.

It is assumed that Cu was doped into the ZnO in the form of ions. EPR measurements could support this assumption. Is there also the possibility of small CuO clusters forming that are within or at the surface of the ZnO lattice? The XRD pattern of the 20% Cu/ZnO sample clearly shows a small reflection at about 40° (Figure 1a). A Rietvield analysis of the XRD pattern with higher Cu loading could help here.

STEM results are presented in Figure 3 and not in Figure 2. Moreover, the quality of Figure 3 and of Figure 9 is very bad. Nothing can be seen from Figure 3 because it its nearly completely black.

The XP spectrum of Zn 2p shows two maxima. However, the spectrum was fitted using only one Zn2+ species. Why?

Were the EIS measurements performed in the dark or under light irradiation?

Reviewer 2 Report

Dear Authors, in your manuscript, the following points should be added/changed to further improve it:

1.      Abstract: What does "7% Cu-ZnO" mean? Is it a quantitative (molar) % or a weight %?

2.      Abstract: Please add information on the size of the nanoparticles obtained.

3.      Keywords: Please add the keyword "zinc oxide".

4.      Introduction: Please indicate/emphasise the novelty of your work.

5.      Introduction: One of the research objectives of the thesis was the solvothermal synthesis of ZnO and doped ZnO. Please add a few sentences describing this method, e.g. about the advantages of solvothermal synthesis. Provide a definition of the method. I believe that the authors had their substantive reasons for choosing exactly this method of synthesis so it is worth signalling this to the readers.  I suggest the authors point out to the reader the relevant review literature on solvothermal synthesis ( DOI: 10.1515/znb-2010-0805, DOI:10.3390/nano10061086).

6.      Results and Discussion - Material characterization: Please add crystallite sizes, calculate crystallite size from XRD results.

7.      Results and Discussion - Material characterization: Please add information that no impurity peaks can be seen in the detection range of the XRD method.

8.      Results and Discussion - Material characterization: Why did the authors provide TEM results for only one sample? There is no explanation to the reader as to what were the reasons and justification for this decision.

9.      Results and Discussion - Material characterization: Please estimate from TEM images the particle size range of 7% Cu-ZnO.

10.  Results and Discussion - Safety test: Why did the authors not include the analysis of Zn2+ ions formed from the dissolution of the sample? There is no discussion of whether Zn2+ ions are toxic to aquatic animals and what the acceptable limits are.   

11.  Results and Discussion - Why did the authors not determine the actual dopant content of all the samples received?

12.  Preparation of Cu-ZnO nanocomposites: Please add Tables with the exact composition of each sample received. How did the authors calculate the dopant content of ZnO?

Round 2

Reviewer 2 Report

Please round the results of the specific area to the first decimal place (Table 2). I do not see the physical sense in giving the result to the fourth decimal place. 

Author Response

Dear Reviewer:

Thank you for your comment concerning our manuscript entitled “Facile synthesis of Cu-doped ZnO nanoparticles for the enhanced photocatalytic disinfection of bacteria and fungi” (Ref. No. molecules-2648706).

Comment: Please round the results of the specific area to the first decimal place (Table 2). I do not see the physical sense in giving the result to the fourth decimal place.

 Response:

We have rounded the results of the specific area to the first decimal place in table 2 as the reviewer suggested in the revised manuscript.